# A Novel Blood–Brain Barrier-Penetrating and Vascular-Targeting Chimeric Peptide Inhibits Glioma Angiogenesis

**DOI:** 10.3390/ijms24108753

**Published:** 2023-05-15

**Authors:** Lu Lu, Longkun Wang, Lin Zhao, Jing Liao, Chunqian Zhao, Xiaohan Xu, Fengshan Wang, Xinke Zhang

**Affiliations:** 1Key Laboratory of Chemical Biology (Ministry of Education), NMPA Key Laboratory for Quality Research and Evaluation of Carbohydrate-Based Medicine, Institute of Biochemical and Biotechnological Drug, School of Pharmaceutical Sciences, Cheeloo College of Medicine, Shandong University, Jinan 250012, China; 2Key Laboratory of Chemical Biology (Ministry of Education), Department of Pharmacology, School of Pharmaceutical Sciences, Cheeloo College of Medicine, Shandong University, Jinan 250012, China

**Keywords:** blood–brain barrier, dual-targeting penetrating peptide, anti-angiogenesis, VEGFR-2, NRP-1

## Abstract

The high vascularization of glioma highlights the potential value of anti-angiogenic therapeutics for glioma treatment. Previously, we designed a novel vascular-targeting and blood–brain barrier (BBB)-penetrating peptide, TAT-AT7, by attaching the cell-penetrating peptide TAT to a vascular-targeting peptide AT7, and we demonstrated that TAT-AT7 could target binding to the vascular endothelial growth factor receptor 2 (VEGFR-2) and Neuropilin-1 (NRP-1), which are both highly expressed in endothelial cells. TAT-AT7 has been proven to be a good targeting peptide which could effectively deliver the secretory endostatin gene to treat glioma via the TAT-AT7-modified polyethyleneimine (PEI) nanocomplex. In the current study, we further explored the molecular binding mechanisms of TAT-AT7 to VEGFR-2 and NRP-1 and its anti-glioma effects. Accordingly, TAT-AT7 was proven to competitively bind to VEGFR-2 and NRP-1 and prevent VEGF-A165 binding to the receptors by the surface plasmon resonance (SPR) assay. TAT-AT7 inhibited endothelial cells’ proliferation, migration, invasion, and tubule formation, as well as promoted endothelial cells’ apoptosis in vitro. Further research revealed that TAT-AT7 inhibited the phosphorylation of VEGFR-2 and its downstream PLC-γ, ERK1/2, SRC, AKT, and FAK kinases. Additionally, TAT-AT7 significantly inhibited angiogenesis of zebrafish embryo. Moreover, TAT-AT7 had a better penetrating ability and could penetrate the BBB into glioma tissue and target glioma neovascularization in an orthotopic U87-glioma-bearing nude mice model, and exhibited the effect of inhibiting glioma growth and angiogenesis. Taken together, the binding and function mechanisms of TAT-AT7 were firstly revealed, and TAT-AT7 was proven to be an effective and promising peptide for the development of anti-angiogenic drugs for targeted treatment of glioma.

## 1. Introduction

Glioma is the most common and aggressive primary brain tumor, characterized by high invasiveness and poor prognosis [1]. Due to its extensive infiltration and rapid spreading to nearby normal brain tissue, glioma is hardly ever completely removed by traditional surgery or radiotherapy [2]. Glioma cells require nutrients and oxygen provided by new blood vessels. Endothelial cell proliferation, migration, and tube formation are key steps in the progression of glioma [3]. In addition, increasing evidence suggests that abnormal angiogenesis contributes to high interstitial fluid pressure and impedes oxygenation within the glioma tissue, which promotes resistance to chemotherapy, radiotherapy, and immunotherapy [4,5]. The proven dependence of glioma growth, colonization, and metastasis on angiogenesis provides a potential rationale of anti-angiogenic strategies for glioma treatment.

Hypoxia in glioma tissue stimulates the expression of the transcription factor, HIF-1α, which triggers the production of excess quantities of pro-angiogenic growth factors. One of the most pivotal pro-angiogenic growth factors involved in glioma angiogenesis is vascular endothelial growth factor A (VEGF-A) [6]. The regulatory mechanism of VEGF-A is mainly mediated via interaction with VEGFR-2, which is highly expressed on glioma and endothelial cells [7]. Upon binding by VEGF-A, VEGFR-2 achieves receptor dimerization and phosphorylation of specific tyrosine residues within its cytoplasmic tail. The activated VEGF-A/VEGFR-2 downstream signal cascades stimulate endothelial cells’ proliferation, survival, migration, and the formation of new vessels, as well as suppress the apoptosis of endothelial cells [8,9]. NRP-1 is a cell-surface transmembrane glycoprotein expressed by endothelial cells and multiple tumor cells [10]. Studies have proven that NRP-1 possesses high affinity to VEGF-A and serves as a co-receptor for VEGFR-2, which enhances the interaction of VEGF-A with VEGFR-2 and significantly increases the angiogenic effects of the downstream signaling pathway [11]. Furthermore, NRP-1 is also found to be able to regulate angiogenesis in a VEGFR-2-independent manner [12,13]. Due to the important roles of VEGF-A165/NRP-1/VEGFR-2 in tumor angiogenesis, we propose that interrupting VEGF-A165 binding to VEGFR-2 or NRP-1 may be a practical method for anti-angiogenic therapy.

Most peptides and proteins are usually functional and efficacious signaling molecules that naturally exist or are synthesized by artificial methods [14]. Compared with proteins, peptides have higher tissue permeability, protease hydrolysis tolerability, and lower production costs. Furthermore, the pharmacological functions of peptides are easier to be improved by modification strategies, whereas their intrinsic properties are rarely affected [15,16,17,18]. Hence, peptides become promising candidates for clinical applications. ATWLPPR (AT7), which was identified by a mutated phage library, was demonstrated to be able to bind to VEGFR-2 and NRP-1 and showed an inhibitory effect on VEGF-induced rabbit corneal angiogenesis [19,20,21,22]. Additionally, the C-terminal arginine (CendR motif) in the AT7 is responsible for increasing tissue and vascular permeability [23]. Therefore, AT7 may be a potential vascular-targeting peptide for glioma therapy.

Intracranial delivery of therapeutic agents is primarily hindered by the BBB [24]. The intercellular tight junction caused by the BBB restricts the passive diffusion of almost all large-molecule and more than 98% of small-molecule drugs from peripheral blood circulation to the brain tissue [25,26]. TAT (RKKRRQRRR), one of the most promising cationic cell-penetrating peptides (CPPs), has been widely applied in drug delivery due to its high efficiency to facilitate the tissue penetration and internalization of cargos and break through a variety of biological barriers, such as cell membranes, nuclear membranes, the BBB, and the blood–retinal barrier (BRB) [27,28,29]. However, similar to most pharmaceutical molecules, TAT shows poor selectivity or specificity to most tissues in vivo. In a previous study, we designed a chimeric peptide TAT-AT7 (RKKRRQRRRCATWLPPR) by coupling TAT to the N-terminus of AT7 using one cysteine as a linker. Compared with TAT or AT7 alone, TAT-AT7 showed higher affinity to VEGFR-2 and NRP-1, and stronger BBB-penetration capabilities. Furthermore, TAT-AT7 exhibited a good targeting ability to the glioma, which could mediate modified PEI gene nanocomplexes (PPTA/pVAXⅠ-En nanocomplex) to overcome the BBB and precisely target glioma tissue, in our previous research [30]. Therefore, based on the characteristics of TAT-AT7, we hypothesized that it cannot only penetrate the BBB and target blood vessels in glioma as a targeting peptide, but it also might directly inhibit glioma angiogenesis and could be developed as a peptide drug to treat glioma.

Hence, in this study, a series of in vitro and in vivo experiments were performed to explore the binding mechanism, properties, and functions of TAT-AT7. Specifically, the effect of TAT-AT7 in competition with VEGF-A165 binding to VEGFR-2 or NRP-1 was evaluated by the SPR assay. The anti-angiogenesis activities of TAT-AT7 in vitro were evaluated by the cell proliferation assay, wound healing assay, transwell invasion assay, tubule formation assay, and the Hoechst and Annexin V-FITC/PI staining assay. Western blotting was used to investigate the molecular mechanism at the cellular level. The in vivo anti-angiogenic activity of TAT-AT7 was firstly evaluated in a zebrafish model. Finally, the glioma vascular-targeting and anti-glioma effects of TAT-AT7 in vivo were evaluated in an intracranial U87-glioma-bearing nude mice model.

## 2. Results

### 2.1. TAT-AT7 Inhibited Proliferation, Migration, Invasion, and Tube Formation of HUVECs In Vitro

AT7, TAT, the mixture of TAT and AT7 (TAT+AT7), and TAT-AT7 groups were set up in our following experiments. The purpose of setting up the TAT+AT7 group was to confirm that the anti-angiogenic activity of TAT-AT7 was not affected by the physical superposition of the TAT and AT7. The anti-proliferative effect was evaluated by the MTT assay. As shown in Figure 1A, TAT-AT7 could significantly inhibit proliferation of HUVECs in a concentration-dependent manner. When the concentration was higher than 40 μmol/L, the inhibition rate of the TAT-AT7 group was significantly higher than that of the AT7, TAT, and TAT+AT7 groups. When the concentration of TAT-AT7 was 320 μmol/L, the inhibition rate of TAT-AT7 on HUVECs was more than 50%.

The effect of TAT-AT7 on the migration and invasion ability of HUVECs was investigated by the wound healing assay and the transwell invasion assay, respectively. As shown in Figure 1B,C, the migration rate of the TAT-AT7 group was significantly lower than that of the control, AT7, TAT, and TAT+AT7 groups at 24 h or 48 h after scratch healing. As shown in Figure 1D,E, the invasion rate of the TAT-AT7 group was significantly lower than that of the other groups. Although the TAT+AT7 group also had a certain inhibitory effect on the invasion of HUVECs, the inhibition rate was significantly lower than that of TAT-AT7.

In order to investigate the effect of TAT-AT7 on the formation of the tubular structure of HUVECs, Matrigel was used to simulate the structure of the basement membrane in the tube formation assay. As shown in Figure 1F,G, compared with the AT7, TAT, and TAT+AT7 groups, the total branch length of the tubular structure of cells treated with TAT-AT7 was significantly reduced. It could significantly inhibit the formation of tubular structures in HUVECs. The above results demonstrate that TAT-AT7 exhibited a significant inhibitory effect on HUVECs’ proliferation, migration, invasion, and tubular structures.

### 2.2. TAT-AT7 Promoted Apoptosis of HUVECs

Hoechst 33,258 staining and Annexin V-FITC/PI staining were used to investigate the effect of TAT-AT7 on apoptosis of HUVECs. Apoptotic cells could be stained to a brighter fluorescence by blue Hoechst 33,258 due to their condensed chromatin. Fluorescent images (Figure 2A) showed that the number of apoptotic cells in the TAT-AT7 group was significantly increased compared to the control, AT7, TAT, and TAT+AT7 groups. Annexin V-FITC/PI staining was then performed by flow cytometric analysis. As shown in Figure 2B,C, the total apoptotic rates of the control, AT7, TAT, and TAT+AT7 groups were (7.17 ± 1.42)%, (7.80 ± 1.50)%, (8.94 ± 3.52)%, and (11.51 ± 1.20)%, respectively, while TAT-AT7 treatment resulted in a significantly increased total apoptotic rate, reaching (29.56 ± 3.07)%. The above results indicate that TAT-AT7 strongly induced HUVECs’ apoptosis, while AT7, TAT, and TAT+AT7 had no obvious effect of promoting apoptosis.

### 2.3. TAT-AT7 Inhibited VEGF-A165 Binding to VEGFR-2 and NRP-1

The anti-angiogenesis effects of TAT-AT7 on HUVECs was demonstrated by the above experiments. In our previous study, we found that TAT-AT7 had high affinity to VEGFR-2 and NRP-1. Due to the lack of consensus signaling domains, NRP-1 mainly serves as a co-receptor for VEGF-A165- and enhanced VEGF-A165-induced VEGFR-2 downstream signal transduction, which promote angiogenesis [11]. Therefore, we speculate that TAT-AT7 may inhibit VEGF-A165 binding to VEGFR-2 and NRP-1, and then block the downstream VEGFR-2 signaling pathway.

To determine the effect of TAT-AT7 in competition with VEGF-A165 binding to VEGFR-2 or NRP-1, the SPR assay was conducted. The whole operation was conducted on a Biacore S200 instrument. The recombinant VEGFR-2 or NRP-1 proteins were firstly immobilized on a CM5 chip by an amine-coupling reaction. VEGF-A165 ligands with gradient concentrations in the absence or presence of different peptides were then flowed over the CM5 chip. The binding of VEGF-A165 ligands to the immobilized recombinant VEGFR-2 or NRP-1 proteins led to an increase of the refractive index on the surface of the chip, which could be converted into the fluctuation of electrical signals in the response (RU). As the concentration of VEGF-A165 was higher, the response value was also higher. However, with the addition of different peptides, the response value decreased to different degrees (Figure 3A and Figure 4A).

As shown in Figure 3B–D, when the concentrations of the 3 peptides were 16 and 64 nmol/L, the binding of VEGF-A165 to VEGFR-2 was not significantly affected. When the concentration of the peptides was increased to 256 and 1000 nmol/L, TAT-AT7 significantly reduced the RU value of the VEGF-A165- and VEGFR-2-binding response at all concentrations, while AT7 and TAT had less inhibitory effects on VEGF-A165 and VEGFR-2 binding than TAT-AT7. As shown in Figure 4B–D, the binding response values of VEGF-A165 and NRP-1 were reduced to different degrees by the three peptides, and the inhibitory effect on the binding of VEGF-A165 and NRP-1 was gradually enhanced with the increase of the peptides’ concentration. When the concentration of TAT-AT7 was 256 nmol/L, the binding curve of VEGF-A165 and NRP-1 tended to be a zero baseline. However, when the concentration of TAT-AT7 increased to 1000 nmol/L, TAT-AT7 completely inhibited the binding of VEGF-A165 and NRP-1. At this time, the same concentration of AT7 and TAT displayed inhibitory effects on the binding of VEGF-A165 and NRP-1, but it did not completely inhibit the binding of them as TAT-AT7 did. The inhibitory effect of TAT was slightly stronger than that of AT7. The above results indicate that TAT-AT7 could significantly inhibit the binding of VEGF-A165 to VEGFR-2 and NRP-1 simultaneously, and it had a stronger inhibitory effect on the binding of VEGF-A165 to NRP-1.

### 2.4. TAT-AT7 Inhibited Phosphorylation of VEGFR-2 and Several Downstream Proteins

Considering the inhibitory effect of VEGF-A165 binding to VEGFR-2 and NRP-1 receptors by TAT-AT7 in the SPR analysis, we speculated that the anti-angiogenic mechanism of TAT-AT7 might be caused by the effects on the downstream signaling pathway of VEGFR-2. Next, we investigated the phosphorylation of VEGFR-2 and some key downstream kinases of VEGFR-2 to explore the molecular mechanism of anti-angiogenesis of TAT-AT7. As shown in Figure 5, TAT-AT7 could reduce the phosphorylation level of VEGFR-2. Furthermore, TAT-AT7 significantly inhibited the phosphorylation level of PLC-γ, ERK1/2, SRC, AKT, and FAK compared with the control, AT7-, TAT-, and TAT+AT7-treated groups. At the same time, TAT-AT7 had no significant effect on the total expression levels of VEGFR-2 and the above kinases.

### 2.5. TAT-AT7 Penetrated into Brain Microvascular Endothelial Cells

Whether TAT-AT7 can cross the BBB is a key to treating glioma. Brain capillary endothelial cells are an essential component of the BBB; hence, the cellular uptake assay in bEnd3 cells (mouse brain microvascular endothelial cells) was conducted to evaluate the penetrating ability of TAT-AT7. The bEnd3 cells were incubated with FITC-AT7, FITC-TAT, FITC-TAT + FITC-AT7, and FITC-TAT-AT7 for 1 h, and then the cellular uptake assay was investigated by fluorescence imaging and flow cytometry. As shown in Figure 6A, FITC-TAT, FITC-TAT+FITC-AT7, and FITC-TAT-AT7 groups presented higher uptake than that of the control and FITC-AT7 groups. The highest uptake was observed in the FITC-TAT-AT7 group, while FITC-AT7 uptake was the lowest. Then, the quantitative analysis was conducted via flow cytometry. The results are shown in Figure 6B,C. The mean fluorescence intensity of the FITC-TAT-AT7 group was about 118, 16, and 11 times higher than that of the FITC-AT7, FITC-TAT, and FITC-TAT + FITC-AT7 groups, which revealed that TAT-AT7 possessed the highest uptake efficiency and penetration capability for bEnd3 cells.

### 2.6. TAT-AT7 Inhibited the Intersegmental Vascular Growth of Zebrafish

The effect of different peptides on the angiogenesis of zebrafish embryos was investigated by observing the green fluorescence intensity of the intersegmental vessel of embryos under a fluorescence microscope. As shown in Figure 7, compared with the control group, the AT7, TAT, and TAT+AT7 groups had no obvious effect on intersegmental vascular growth, while TAT-AT7 could significantly inhibit the generation of the zebrafish embryo intersegmental vessel, which had a strong anti-angiogenesis effect in vivo.

### 2.7. Safety Evaluation of TAT-AT7 In Vivo

The hemolysis assay was used to investigate the safety of TAT-AT7 in vivo. As shown in Figure 8, AT7, TAT, TAT+AT7, and TAT-AT7 did not cause obvious hemolysis in the concentration range of 2.5–640 μmol/L, and the hemolysis rate was lower than 5%. The results indicated that the hemolysis of AT7, TAT, and TAT-AT7 was weak and had good biosafety, and could be further used for an in vivo experimental study.

### 2.8. TAT-AT7 Showed High Penetration into Glioma and Vascular-Targeting Capability In Vivo

To investigate the glioma tissue distribution and vascular-targeting of TAT-AT7, we conducted an immunohistochemical analysis of blood vessel marker CD31-staining in orthotopic U87-mCherry-luc glioma-bearing nude mice. Figure 9 shows the brain glioma distribution and accumulation after intravenous tail injection with FITC-AT7, FITC-TAT, and FITC-TAT-AT7. The green fluorescence intensity in the TAT-AT7 group was significantly higher than that in the AT7 and TAT groups, suggesting a distinct increase in glioma tissue permeability and accumulation of TAT-AT7. Meanwhile, TAT-AT7 was obviously co-localized with glioma blood vessels, which were immunohistochemically stained for CD31. There was very weak green fluorescence in the AT7 group, mainly due to the poor ability of AT7 to penetrate the BBB. The above results were consistent with the trend of the cellular uptake experiment in bEnd3 cells and indicated that TAT-AT7 possessed excellent BBB crossing, glioma tissue penetration, and specific vascular-targeting abilities in vivo.

### 2.9. TAT-AT7 Inhibited the Growth of Glioma in Nude Mice

The anti-glioma effect of TAT-AT7 in vivo was evaluated in an intracranial U87-mCherry-luc glioma-bearing nude mice model. The luciferase in U87-mCherry-luc cells can react with the D-Luciferin potassium salt and generate bioluminescence, which can be monitored by the IVIS spectrum imaging system. The bioluminescence intensity is proportional to glioma growth. U87-mCherry-luc glioma-bearing nude mice were randomly divided into five groups. The control group was treated with normal saline.

It was clear that TAT-AT7 significantly suppressed glioma growth on days 11 and 14 compared with the other groups treated with saline, AT7, TAT, and TAT+AT7 (Figure 10A,B). In contrast, the glioma of the AT7, TAT, and TAT+AT7 groups still grew at a fast rate, which had no significant difference compared to the control group. The results of the body weight change in nude mice are shown in Figure 10C. As glioma infiltrates and grows in the brain, it presents a high degree of malignancy. The body weight of nude mice showed a downward trend. However, the rate of weight loss in the TAT-AT7 group was slower than that in the normal saline group, and the physiological status of the nude mice in the TAT-AT7 group was better at the end of administration on the 14th day, indicating that TAT-AT7 had less toxicity and better safety for nude mice. TUNEL staining was used to determine the existence of apoptosis by observing the disruption of nuclear DNA in apoptosis. CD31 is a representative vascular marker which can be used to highlight glioma vessels. Therefore, CD31 immunohistochemical staining of glioma tissue was conducted. As illustrated in Figure 10D, the number of apoptotic cells in the TAT-AT7 group was significantly higher than that in the saline, AT7, TAT, and TAT+AT7 groups, while the blood vessel density was significantly lower than that in the other groups. The above results demonstrate that TAT-AT7 could significantly inhibit the growth of glioma in nude mice, induce apoptosis of glioma cells, and inhibit angiogenesis of glioma.

## 3. Discussion

In most research, AT7 or TAT are usually used as targeting or penetrating ligands to modify drug delivery systems for the treatment of different diseases [28,31,32,33]. We raised an idea to construct the chimeric peptide TAT-AT7 and directly evaluated its biological activities in vitro and in vivo. TAT-AT7 surprisingly displayed remarkable inhibitory effects on endothelial cells’ proliferation, migration invasion, and tube formation compared to AT7, TAT, and TAT+AT7, respectively. TAT-AT7 also showed the unique effect of promoting endothelial cells’ apoptosis. In addition, TAT-AT7 inhibited the intersegmental vascular growth of zebrafish in vivo.

The BBB prevents most drugs from reaching the brain [25]. Thus, evaluating whether TAT-AT7 can successfully cross the BBB is the prerequisite for studying the anti-glioma effect of TAT-AT7. Cellular uptake results (Figure 6) showed that the existence of the AT7 sequence in TAT-AT7 did not weaken the cell-penetrating ability of the TAT peptide to bEnd3 cells. On the contrary, the cell uptake ability of TAT-AT7 was much stronger than that of TAT or TAT+AT7, which implied that the conjunction of AT7 to TAT significantly improved the selective penetration ability of TAT to the brain microvascular endothelial cells. The immunofluorescence analysis of frozen intracranial U87 glioma tissue sections from nude mice revealed that TAT-AT7 exhibited higher accumulation and deeper distribution in the glioma region after intravenous injection compared to TAT and TAT+AT7, while AT7 was unable to achieve obvious accumulation in glioma. Meanwhile, TAT-AT7 showed obvious co-localization with the blood vessel marker CD31, indicating that TAT-AT7 could specifically bind to neovascular glioma. The stronger BBB penetration, high glioma tissue accumulation, and vascular-targeting ability of TAT-AT7 provided more possibilities to enhance the anti-glioma and anti-angiogenesis effects and reduce the toxicity to normal tissues. In addition, although AT7 is a vascular-targeting peptide, the limited BBB penetration might hinder its potency. The BBB is a highly complex structure which regulates contacts between the peripheral circulation and the central nervous system. We speculated that TAT-AT7 can penetrate the BBB mainly through adsorption-mediated transcytosis via electrostatic interaction of the cationic TAT-AT7 with the negatively charged BBB [34,35]. On the other hand, TAT-AT7 penetrated the BBB more efficiently via the specific binding of TAT-AT7 to VEGFR-2 and NRP-1, both of which were overexpressed on the brain capillary endothelial cells of BBB. In addition, the interaction of TAT-AT7 with the NRP-1 also increased vascular permeability and tissue internalization. The in vivo anti-glioma efficacy of TAT-AT7 was evaluated in nude mice bearing intracranial U87-luc glioma. Administration of TAT-AT7 led to a significant inhibitory effect on glioma growth and angiogenesis compared to AT7, TAT, and TAT+AT7. The hemolysis assay and the changes of body weight in the nude mice results revealed that TAT-AT7 had good safety in vivo.

Different from traditional functional applications of TAT that are mainly used as a drug delivery tool, our present study found that TAT-AT7 presented more potent BBB penetration and anti-angiogenic effects after conjugating TAT to AT7. Then, we further performed a preliminary exploration about the potential mechanism of action mediated by TAT-AT7. Through the SPR assay, we first provided evidence that TAT-AT7 potently inhibited the VEGF-A165 binding to VEGFR-2 and NRP-1, simultaneously. A previous study found that the four C-terminal residues “LPPR” are essential for the inhibitory effect of AT7 on VEGF-A165 binding to NRP-1 [21]. Zachary et al. reported that some cysteine-rich HIV-1 Tat-derived peptides (such as KRRQRRRPPQGNQAHQDPLP, SYGRKKRRQRRR, KKRRQRRRPPQG) displayed potency in disturbing the cross-linking of VEGF to NRP-1 [36]. Erkki’s research proposed that peptides containing a C-terminal arginine residue or an internal R/KXXR/K sequence could interact with NRP-1 [37]. TAT-AT7 retained the C-terminal residues LPPR and added the N-terminal sequence KKRRQRRR, which might explain why the inhibitory effect of TAT-AT7 on VEGF-A165 binding to NRP-1 was much higher than that of AT7 and TAT. Besides, our results showed that TAT exhibited a stronger competitive binding ability to NRP-1 than AT7, which suggested that TAT might possess a higher affinity to NRP-1 than AT7. We will carry out further studies to demonstrate the targeting property of TAT to NRP-1 in the future.

Upon VEGF-A165 binding, the dimerization of VEGFR-2 results in the phosphorylation of specific tyrosine residues within its cytoplasmic tail. NRP-1 functions as a co-receptor to form a VEGF-A165-dependent complex with VEGFR-2 and amplifies the activity of a series of downstream transduction signaling pathways of VEGFR-2, including PLC-γ, ERK1/2, SRC, AKT, FAK, etc. [12,38,39,40,41]. Therefore, the blocking of VEGF-A165 binding to VEGFR-2 or NRP-1 will ultimately lead to changes in the downstream transduction signaling pathways of VEGFR-2. Western blotting results indicated that treatment with TAT-AT7 significantly downregulated VEGF-induced phosphorylation levels of VEGFR-2, PLC-γ, ERK1/2, SRC, AKT, and FAK in HUVECs, compared to the AT7-, TAT-, or TAT+AT7-treated groups. Considering that the PLC-γ/ERK1/2 signaling is a crucial pathway for regulating proliferation of endothelial cells, the activation of FAK is closely related to cell migration, and the SRC/AKT signaling facilitates the apoptosis of endothelial cells, we deduced that the antiangiogenic molecular mechanisms were that TAT-AT7 inhibited proliferation of endothelial cells through the VEGFR2/PLC-γ/ERK signaling pathway, suppressed migration of endothelial cells via the VEGFR2/FAK signaling pathways, and facilitated the apoptosis of endothelial cells by downregulating the VEGFR2/SRC/AKT signaling pathway. However, we could not exclude the possibility that TAT-AT7 suppressed angiogenesis through other mechanisms, such as inhibiting the VEGFR-2-independent NRP-1 signaling pathway in HUVECs. Further studies will need to validate the exact mechanism of action and effectiveness of TAT-AT7 in other in vitro and in vivo models.

## 4. Materials and Methods

### 4.1. Reagents

RKKRRQRRR (TAT), ATWLPPR (AT7), and RKKRRQRRRCATWLPPR (TAT-AT7) were synthesized from China Peptides Co., Ltd. (Shanghai, China). The Hoechst 33,258 staining kit, Annexin V-FITC/PI apoptosis detection kit, and BCA protein assay kit were obtained from Beyotime Biotechnology (Nantong, China). The total protein extraction kit was from BestBio (Shanghai, China). The CM5 sensor chip was obtained from GE Healthcare Life Sciences (Uppsala, Sweden). Purified recombinant human VEGFR-2 and NRP-1 receptor proteins were purchased from R&D Systems (Minneapolis, MN, USA). Recombinant human endothelial growth factor-A165 (VEGF-A165) was obtained from PeproTech (Rocky Hill, NJ, USA). Anti-ERK1/2, anti-p-ERK1/2, anti-p-VEGFR-2, anti-AKT, anti-p-AKT, anti-p-SRC, anti-p-FAK, anti-p38 MAPK, anti-p-p38 MAPK, anti-p-PLC-γ, and anti-GAPDH antibodies were purchased from Cell Signaling Technology (Danvers, MA, USA). Anti-VEGFR-2, anti-FAK, and anti-SRC were purchased from Proteintech Group, Inc (Wuhan, China). F-12K medium, heparin, and endothelial cell growth supplement (ECGS) were from Macgene (Beijing, China). DMEM medium and fetal bovine serum (FBS) were obtained from Gibco^®^, Life Technologies (Carlsbad, CA, USA). Matrigel was purchased from BD Biosciences (San Jose, CA, USA). The 96-well and 6-well plates were bought from Corning Inc. (New York, NY, USA). All other reagents were of the highest commercial grade available.

### 4.2. Cell Lines and Cell Culture

Human umbilical vein endothelial cells (HUVECs) were obtained from ATCC (Manassas, VA). The cells were cultured in F-12K medium supplemented with 10% FBS, 100 μg/mL of heparin, and 50 μg/mL of ECGS under a 5% CO_2_ atmosphere at 37 °C. U87-mCherry-luc cells (human glioma cells) were obtained from Shanghai Sciencelight Biology Science and Technology Co. (Shanghai, China). Mouse brain microvascular endothelial cells (bEnd3 cells) were purchased from Procell Life Science and Technology Co., Ltd. (Wuhan, China). U87-mCherry-luc and bEnd3 cells were cultured in DMEM medium supplemented with 1% penicillin and streptomycin and 10% FBS, under a 5% CO_2_ atmosphere at 37 °C.

### 4.3. Animal Model

Female nude mice (4–6 weeks old) were purchased from the Jinan Pengyue Experimental Animal Breeding Co., Ltd (Jinan, China). Mice were raised under specific pathogen-free (SPF) conditions. All animal procedures were approved by the Ethics Committee of Shandong University and performed in accordance with the Shandong Council on Animal Care.

### 4.4. SPR Analysis

SPR analysis is an optical technology to be used for studying ligand–receptor interactions. The SPR assay was performed using the Biacore S200 instrument (GE Healthcare, Uppsala, Sweden). Firstly, the CM5 sensor chip was activated with EDC/NHS. Then, the recombinant VEGFR-2 and NRP-1 proteins were immobilized on the surface of the sensor chip according to the operating instructions of amine coupling. VEGF-A165 was diluted to 8, 4, 2, 1, and 0.5 nmol/L in different concentrations of peptide solution dissolved by HBS-EP buffer (0, 0.016, 0.064, 0.256, and 1 μmol/L, respectively). The Kinetics/Affinity program was selected, and the automatic continuous injection program was set. The injection flow rate was 30 μL/min, the injection time was 120 s, and the dissociation time was 220 s. At the same time, zero-concentration injection wells and 0.5 nmol/L repeat-concentration injection wells of VEGF-A165 were set. At the end of each injection, the channels were flushed with glycine-HCl solution for 30 s until response values were returned to baseline.

### 4.5. In Vitro Cellular Uptake Experiment

In order to explore whether the permeability of TAT-AT7 to bEnd3 cells could be enhanced, a cellular uptake experiment was conducted. The bEnd3 cells were seeded into 6-well culture plates at 2 × 10^5^ cells/well. When 80% confluence was reached, the cells were treated with 20 μM of FITC-AT7, FITC-TAT, FITC-AT7 and FITC-TAT (FITC-AT7 + FITC-TAT), and FITC-TAT-AT7 diluted in DMEM medium for 1 h (37 °C, 5% CO_2_). The green fluorescence images of bEnd3 cells were visualized under an inverted fluorescence microscope and the fluorescence intensity was determined by flow cytometry.

### 4.6. Cell Proliferation Assay

Briefly, HUVECs were seeded into 96-well plates (5 × 10^3^ cells/well) and allowed to attach overnight. The cells were treated with AT7, TAT, the mixture of TAT and AT7 (TAT+AT7), and TAT-AT7 at a series of concentrations (20, 40, 80, 160, 320 μmol/L) in the presence of VEGF-A165 (30 ng/mL) for 48 h. Then, 20 µL of MTT solution was added to each well and further cultured for 4 h under dark conditions. After removing the medium, 150 µL of DMSO was added to dissolve formazan crystals. The 96-well plates were shaken at a low speed for 10 min. The absorbance was measured at 490 nm using a Microplate Reader (Bio-Rad 680, Hercules, CA, USA).

### 4.7. Wound Healing Assay

HUVECs were firstly seeded into 6-well culture plates (3 × 10^5^ cells/well). Following overnight incubation, the monolayer HUVECs were scraped by pipette tips of 200 μL and washed by cold PBS. AT7, TAT, TAT+AT7, and TAT-AT7 (160 μmol/L) in the presence of VEGF-A165 (30 ng/mL) were added to the plates. The cells were incubated for another 48 h. Images were captured at 0 h, 24 h, and 48 h using an inverted microscope. The cell migration rate was counted by the ImageJ software.

### 4.8. Transwell Invasion Assay

Matrigel was diluted in serum-free F-12K medium (dilution ratio 1:5) and was coated onto each transwell chamber and incubated at 37 °C for 2 h. HUVECs were seeded in the upper compartment of the transwell chamber at a density of 2 × 10^4^ cells/well. F-12K medium (containing 20% serum) was added to the lower chamber. HUVECs were incubated with AT7, TAT, TAT+AT7, and TAT-AT7 (160 μmol/L) in the presence of VEGF-A165 (30 ng/mL) in F-12K medium supplemented with 10% FBS and were incubated at 37 °C for 24 h. Then, the medium was aspirated, and the chambers were washed with PBS and placed in a fixative for 30 min, stained with 0.1% crystal violet for 30 min, washed with PBS, and dried. An inverted fluorescence microscope was used to photograph the chambers, and the number of invaded cells was counted. The cells that migrated to the bottom of the chamber were photographed with an inverted microscope.

### 4.9. Tube Formation Assay

Matrigel was added to 96-well plates with 60 min of incubation at 37 °C for solidification. HUVECs were seeded into the Matrigel-coated plates (3 × 10^4^ cells/well) and incubated with AT7, TAT, TAT+AT7, and TAT-AT7 (160 μmol/L) in the presence of VEGF-A165 (30 ng/mL) for 6 h. The tubular structures were observed under an inverted microscope and the total branching lengths were calculated by the ImageJ V 1.8.0 software.

### 4.10. Hoechst Staining Assay

HUVECs were seeded into 6-well culture plates (3 × 10^5^ cells/well). After overnight incubation, the cells were washed with PBS and treated with AT7, TAT, TAT+AT7, and TAT-AT7 (320 μmol/L) for 24 h. After discarding the supernatant and washing with cold PBS, the cells were fixed in 4% paraformaldehyde and stained with Hoechst 33,258 for 10 min. Then, the cells were washed with cold PBS and observed using an inverted fluorescence microscope.

### 4.11. Annexin V-FITC/PI Staining Assay

HUVECs were treated with different peptides (the methods were the same as the Hoechst staining). After 24 h, cells were collected by trypsin digestion and washed with PBS. Annexin V-FITC/PI staining was conducted according to the manufacturer’s instructions for the Apoptosis Detection Kit. Briefly, 195 μL of Annexin V-FITC/PI dilution buffer was added to resuspend the cells. Then, 5 μL of Annexin V-FITC and 10 μL of PI staining solutions were added and gently mixed with the cells. After incubation at room temperature in the dark for 10 min, the apoptotic cells were detected by flow cytometry.

### 4.12. Western Blotting Analysis

For Western blotting analysis, HUVECs were seeded into 6-well culture plates at 3 × 10^5^ cells/well and incubated overnight. After washing with PBS, cells were treated with AT7, TAT, TAT+AT7, and TAT-AT7 (320 μmol/L) in the presence of VEGF-A165 (30 ng/mL) for 90 min. Two control groups (F-12K medium in the absence or presence of VEGF-A165) were set. The cells were washed with cold PBS and lysed with lysis buffer (protease and phosphatase inhibitors) on the ice for 30 min, and then centrifuged at 4 °C for 15 min at 14,000 r/min. The supernatant was collected, and its total protein concentration was determined by the BCA protein assay kit. Equal amounts of proteins were separated by 10% SDS-PAGE gels and transferred to polyvinylidene difluoride (PVDF) membranes. After blocking with 5% non-fat milk for 1 h, the membranes were incubated with specific antibodies at 4 °C overnight. Then, the membranes were washed with TBST (Tris-Buffered Saline with 0.1% Tween 20) and incubated with horseradish peroxidase (HRP)-conjugated secondary antibody for 1 h at room temperature. The proteins on the membranes were visualized with enhanced chemiluminescence agents.

### 4.13. The Zebrafish Experiment

Female and male transgenic zebrafish were reared in alternating environments of 14 h light and 10 h dark, respectively. After a certain period, healthy, sexually mature female and male zebrafish were mated in a 1:1 ratio. After 24 h, the fertilized eggs were removed, disinfected, and washed, and then placed in embryo culture water and cultured in an incubator at 28 °C for 24 h. The fertilized eggs were washed with chain enzyme protease E solution (1 mg/mL) to remove the egg membrane on the surface and placed under a microscope to observe and select the normally developing zebrafish embryos, and then placed into a 24-well plate. AT7, TAT, TAT+AT7, and TAT-AT7 at a concentration of 25 µmol/L were added to the plates, respectively. The control group was added with the same volume of embryo culture water, and then placed in an incubator at 28 °C for 24 h. Zebrafish embryos were randomly removed from each group and observed under a fluorescence microscope. The total length of zebrafish internode vessels was calculated and counted.

### 4.14. Hemolytic Experiment

The fresh blood of mice was centrifuged at 2000 r/min for 10 min, and the upper layer of serum was discarded, then mixed with normal saline and centrifuged at 2000 r/min for 10 min, repeated three times, and the supernatant was removed to obtain the lower layer of red blood cells. Then, 1 mL of red blood cells was diluted to 50 mL with normal saline. AT7, TAT, TAT+AT7, and TAT-AT7 solutions diluted with normal saline (1280, 640, 320, 160, 80, 40, 20, 10, 5 μmol/L, respectively) were added to equal volumes of the red blood cell suspension, respectively. The samples were incubated at 37 °C for 1 h, centrifuged at 2000 r/min for 10 min, and photographed. Light absorption of samples was measured at 540 nm on a microplate reader.

Cell hemolysis rate (%) = (OD experimental group − negative control group)/(OD positive control group − negative control group) × 100%.

### 4.15. The Vascular-Targeting Assay of TAT-AT7 In Vivo

The orthotopic U87-mCherry-luc glioma-bearing nude mice model was firstly established. Female nude mice were anesthetized by isoflurane and fixed on a ventricle stereotaxic instrument. U87-mCherry-luc cells were collected after trypsin digestion and then injected into the right striatum of the brain of nude mice. On the 4th day after the surgery, the bioluminescence of the glioma tissue was detected by the IVIS kinetic imaging system (Lumina II; Caliper, MA, USA). The mice were randomly divided into four groups according to the bioluminescence intensity (*n* = 3). FITC-AT7, FITC-TAT, and FITC-TAT-AT7 (30 mg/Kg) were injected into mice via the tail vein on day 8. The control group was only injected with normal saline. After 1 h, the mice were sacrificed, and the brains were collected and fixed with 4% paraformaldehyde for 24 h. Then, the fixed brains were dehydrated by 15% and 30% sucrose, successively, frozen with O.C.T. embedding reagent, and then cut into frozen brain tissue slices. Then, sections of slices were incubated with rabbit anti-murine CD31 antibody, followed by treatment with Cy5.5-labeled goat anti-rabbit IgG secondary antibody. The slices were further stained with DAPI for 5 min and observed using a confocal microscope.

### 4.16. In Vivo Anti-Glioma Study

The orthotopic U87-mCherry-luc glioma-bearing nude mice were randomly assigned to 6 groups (*n* = 5) according to the bioluminescence intensity of glioma measured on day 4. The mice in five of the groups were intravenously injected with normal saline, TAT, AT7, TAT+AT7, and TAT-AT7 (20 mg/kg/day), respectively, every day for 11 days. The bioluminescence intensity was measured on days 4, 7, 11, and 14, respectively. Nude mice were weighed daily. On day 15, the mice were sacrificed, and the glioma-bearing brains were harvested. The brain tissues were fixed with 4.0% paraformaldehyde, embedded in paraffin, and cut into slices. The TUNEL staining and CD31 immunohistochemical staining were conducted according to standard histological procedures.

### 4.17. Statistical Analysis

The data were expressed as mean ± standard deviation (SD). All statistical analyses were determined by Student’s *t* test or one-way ANOVA (GraphPad Prism 6.0). The value of *p* < 0.05 was considered as statistically significant.

## 5. Conclusions

In conclusion, we demonstrated that the chimeric peptide TAT-AT7 had a targeted binding ability to VEGFR-2 and NRP-1. TAT-AT7 could inhibit angiogenesis by downregulating phosphorylation of VEGFR2-related signaling pathways. Moreover, TAT-AT7 showed a strong penetrating ability into glioma tissue and a glioma neovascularization-targeting property. In vivo experiments showed that TAT-AT7 could effectively inhibit glioma growth and angiogenesis. In conclusion, TAT-AT7 may present a new strategy for targeted therapy of glioma.

## Figures and Tables

**Figure 1 ijms-24-08753-f001:**
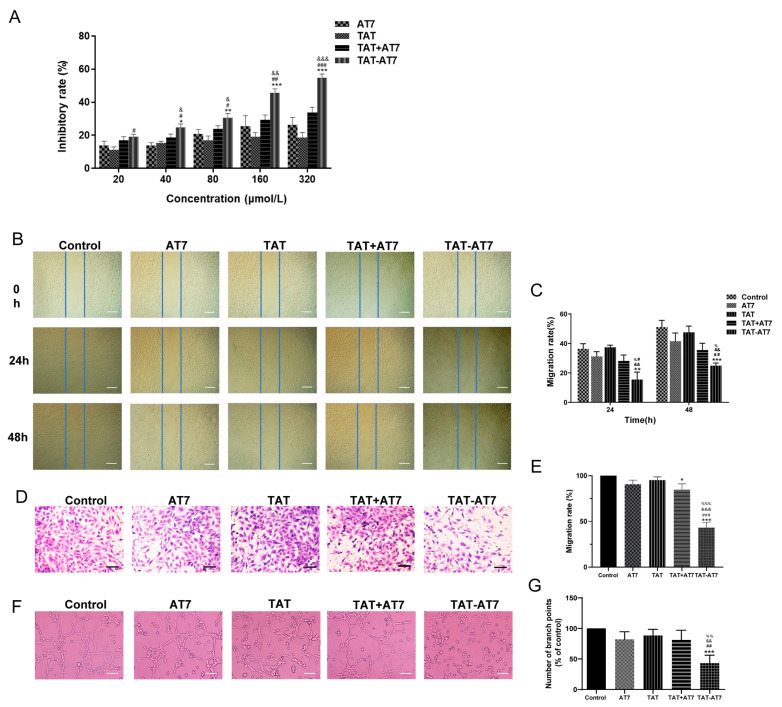
Effects of TAT-AT7 on angiogenesis in vitro. (**A**) Inhibitory effect of AT7, TAT, TAT+AT7, and TAT-AT7 on HUVECs’ proliferation determined by the MTT assay. Each data point represents the mean ± standard deviation. *n* = 3, * *p* < 0.05, ** *p* < 0.01, *** *p* < 0.001 vs. AT7 group; ^#^
*p* < 0.05, ^##^
*p* < 0.01, ^###^
*p* < 0.001 vs. TAT group; ^&^
*p* < 0.05, ^&&^
*p* < 0.01, ^&&&^
*p* < 0.001 vs. TAT+AT7 group. (**B**) AT7, TAT, TAT+AT7, and TAT-AT7 were incubated with HUVECs for 48 h, and the cell migration was photographed under an inverted microscope at 0 h, 24 h, and 48 h, respectively. Bars represent 200 μm. (**C**) Statistical migration rates. (**D**) Different groups of peptides were incubated with HUVECs for 24 h, and the cell invasion was photographed under an inverted microscope. Bars represent 60 μm. (**E**) Statistical invasion rates. (**F**) HUVECs were treated with AT7, TAT, TAT+AT7, and TAT-AT7 for 6 h, and the tubular structure formation of HUVECs was photographed under an inverted microscope. Bars represent 50 μm. (**G**) Statistical total branching length of tubular structures. Each data point represents the mean ± standard deviation. *n* = 3, * *p* < 0.05, ** *p* < 0.01, *** *p* < 0.001 vs. control group; ^#^
*p* < 0.05, ^##^
*p* < 0.01, ^###^
*p* < 0.001 vs. AT7 group; ^&&^
*p* < 0.01, ^&&&^
*p* < 0.001 vs. TAT group; ^%^
*p* < 0.05, ^%%^
*p* < 0.01, ^%%%^
*p* < 0.001 vs. TAT+AT7 group.

**Figure 2 ijms-24-08753-f002:**
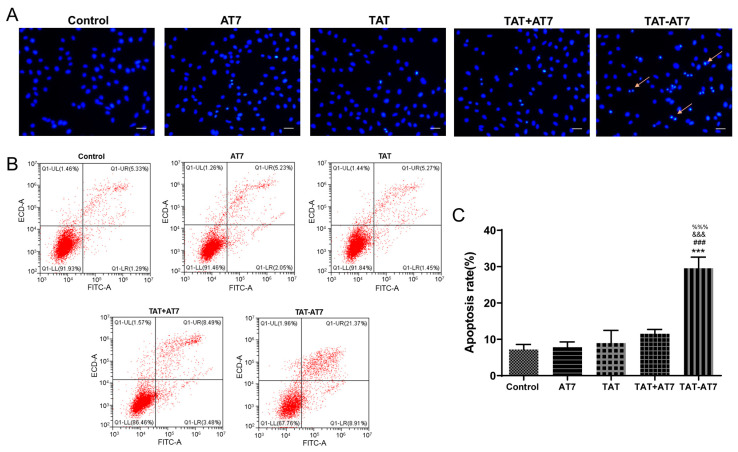
Effect of TAT-AT7 on apoptosis of HUVECs. (**A**) Fluorescence images of HUVECs stained with Hoechst 33,258. Arrows represent the apoptotic cells. Bars represent 50 μm. (**B**) Flow cytometry analysis of apoptosis using Annexin V and PI staining. The late apoptosis cells were indicated as Q1-UR, and the early apoptosis cells were indicated as Q1-LR. (**C**) The total rate of apoptosis of cells (Q1-UR and Q1-LR). Each data point represents the mean ± standard deviation. *n* = 3, *** *p* < 0.001 vs. control group; ^###^
*p* < 0.001 vs. AT7 group; ^&&&^
*p* < 0.001 vs. TAT group; ^%%%^
*p* < 0.001 vs. TAT+AT7 group.

**Figure 3 ijms-24-08753-f003:**
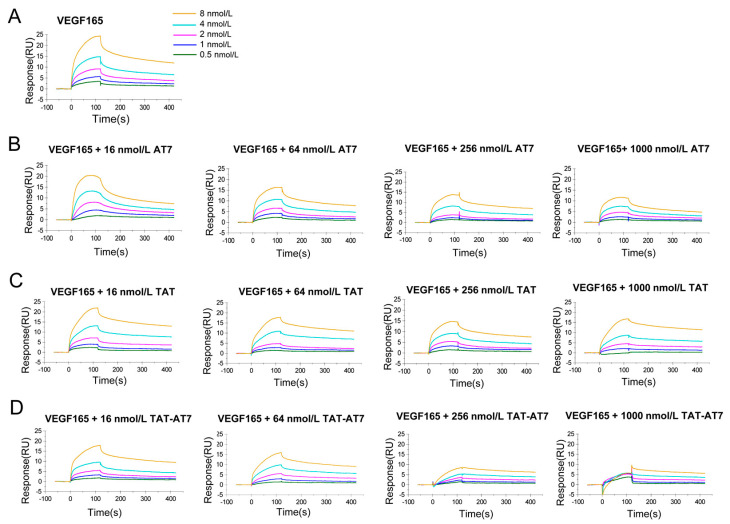
The effects of TAT-AT7 in competition with VEGF-A165 binding to VEGFR-2. (**A**) The changes in RU values of VEGF-A165 binding to VEGFR-2. (**B**–**D**) The changes in RU values of VEGF-A165 binding to VEGFR-2 in AT7, TAT, and TAT-AT7 solutions at concentrations of 16, 64, 256, and 1000 nmol/L, respectively.

**Figure 4 ijms-24-08753-f004:**
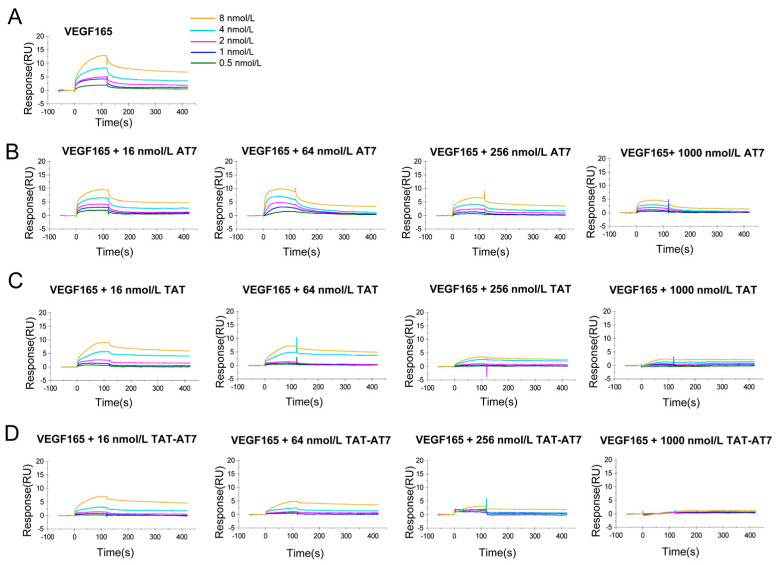
The effects of TAT-AT7 in competition with VEGF-A165 binding to NRP-1. (**A**) The changes in RU values of VEGF-A165 binding to NRP-1. (**B**–**D**) The changes in RU values of VEGF-A165 binding to NRP-1 in AT7, TAT, and TAT-AT7 solutions at concentrations of 16, 64, 256, and 1000 nmol/L, respectively.

**Figure 5 ijms-24-08753-f005:**
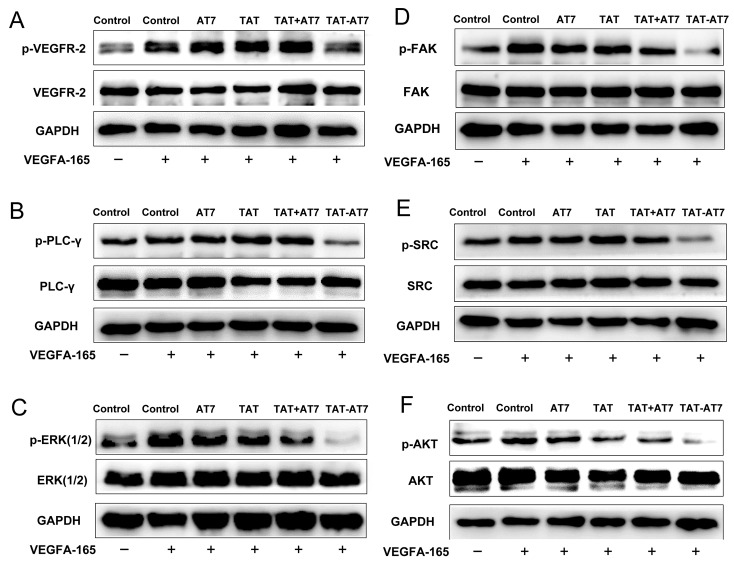
TAT-AT7 inhibited the VEGFR-2 downstream signaling pathway. The expression levels of VEGFR-2 and *p*-VEGFR-2 (**A**), PLC-γ and p-PLC-γ (**B**), ERK1/2 and p-ERK1/2 (**C**), FAK and p-FAK (**D**), SRC and p-SRC (**E**), and AKT and p-AKT (**F**) in HUVECs treated with AT7, TAT, TAT+AT7, and TAT-AT7 in the presence of VEGF-A165 (30 ng/mL), determined by Western blot. *n* = 3.

**Figure 6 ijms-24-08753-f006:**
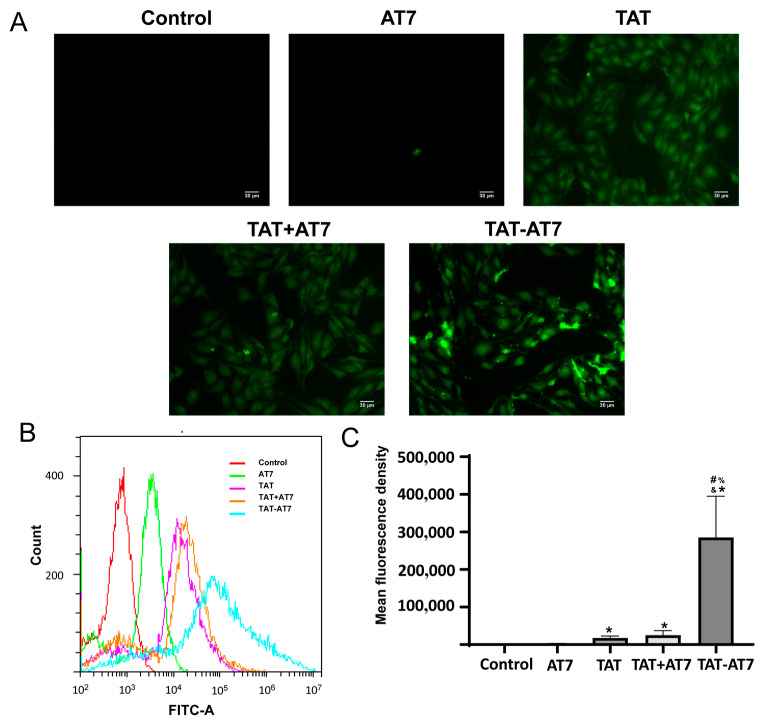
The cellular uptake of TAT-AT7 by bEnd3 cells. (**A**) The FITC-labeled AT7, TAT, and TAT-AT7 were incubated with bEnd3 cells for 1 h, and the fluorescent uptake of peptides by bEnd3 cells was determined under a fluorescence microscope. Bars represent 30 μm. (**B**) Flow cytometric histogram profiles of fluorescence intensity. (**C**) The statistical mean fluorescence intensity. Each data point represents the mean ± standard deviation. *n* = 3, * *p* < 0.05 vs. control group; ^#^
*p* < 0.05 vs. AT7 group; ^&^
*p* < 0.05 vs. TAT group; ^%^
*p* < 0.05 vs. TAT+AT7 group.

**Figure 7 ijms-24-08753-f007:**
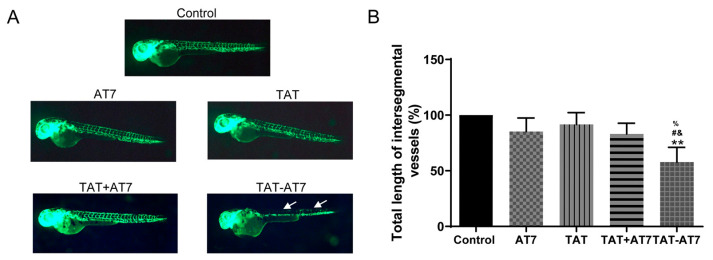
TAT-AT7 inhibited the intersegmental vascular growth of zebrafish. Zebrafish embryos were treated with AT7, TAT, TAT+AT7, and TAT-AT7. After 24 h, the angiogenesis of zebrafish was observed under a fluorescence microscope. (**A**) Photographs of intersegmental vascular growth of zebrafish. (**B**) Statistical total length of intersegmental vessels. Each data point represents the mean ± standard deviation. *n* = 3, ** *p* < 0.01 vs. control group; ^#^
*p* < 0.05 vs. AT7 group; ^&^
*p* < 0.05 vs. TAT group; ^%^
*p* < 0.05 vs. TAT+AT7 group.

**Figure 8 ijms-24-08753-f008:**
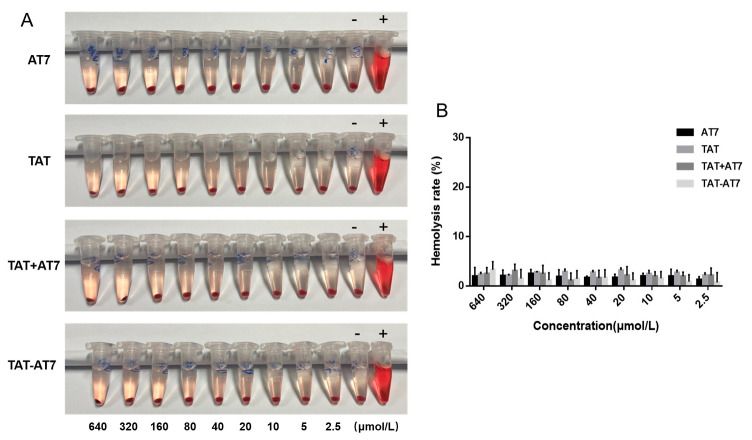
Hemolysis assay results of TAT-AT7. Different concentrations of AT7, TAT, TAT+AT7, and TAT-AT7 were incubated with red blood cells, and the hemolysis rate was measured after 1 h. The concentrations of peptides were 640, 320, 160, 80, 40, 20, 10, 5, and 2.5 μmol/L, from left to right, where “-” indicates the negative control group, and “+” indicates the positive control group. (**A**) Photographic observation of hemolysis. (**B**) Statistical results of the hemolysis rate. Each data point represents the mean ± standard deviation. *n* = 3.

**Figure 9 ijms-24-08753-f009:**
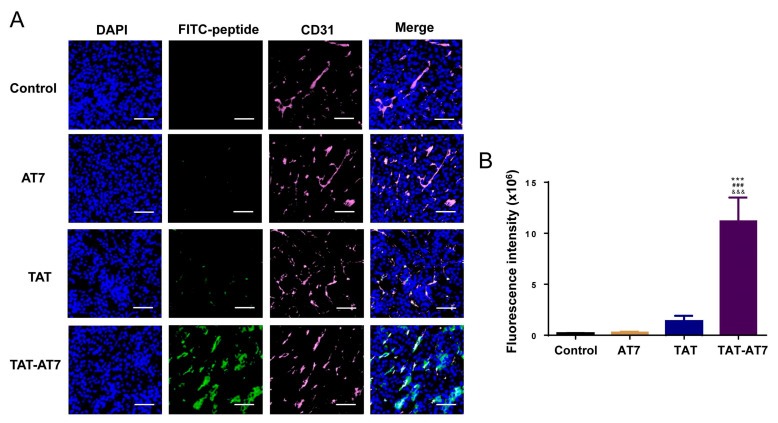
TAT-AT7 targeted glioma blood vessels in the intracranial glioma tissue of nude mice. FITC-labeled AT7, TAT, and TAT-AT7 were injected into intracranial U87-mCherry-luc glioma-bearing mice via the tail vein. After 1 h, the glioma tissues of nude mice were taken for frozen sectioning. (**A**) The fluorescence distribution of tissues was observed under a panoramic scanning microscope after CD31 immunofluorescence staining. Blue fluorescence: nucleus stained with DAPI, green fluorescence: FITC-labeled peptide, pink fluorescence: CD31 stained with Cy5.5. Bars represent 200 μm. (**B**) Statistical analysis results of FITC fluorescence intensity. Each data point represents the mean ± standard deviation. *n* = 3, *** *p* < 0.001 vs. control group; ^###^
*p* < 0.05 vs. AT7 group; ^&&&^
*p* < 0.05 vs. TAT group.

**Figure 10 ijms-24-08753-f010:**
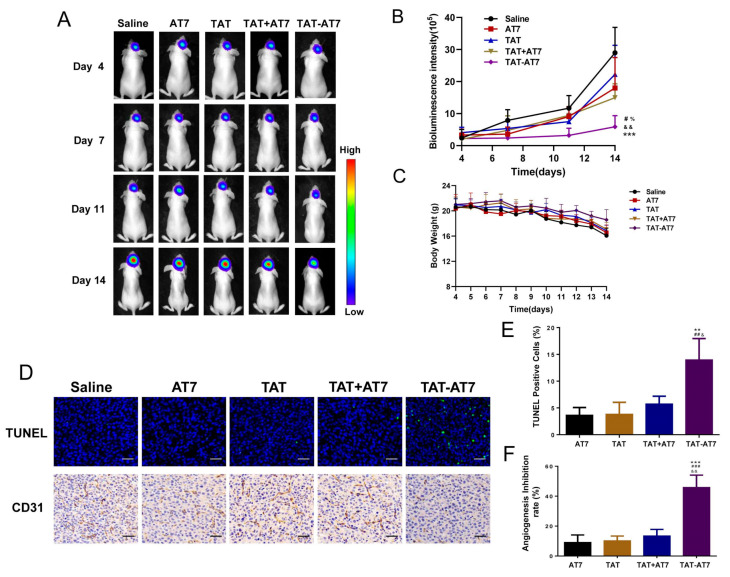
TAT-AT7 inhibited the growth of glioma in nude mice. Normal saline, AT7, TAT, TAT+AT7, and TAT-AT7 were injected into intracranial U87-mCherry-luc glioma-bearing mice daily via the tail vein for 11 consecutive days. (**A**) The bioluminescence images were visualized by the IVIS imaging system on days 4, 7, 11, and 14. Each data point represents the mean ± standard deviation. *n* = 5, *** *p* < 0.001 vs. control group; ^#^
*p* < 0.05 vs. AT7 group; ^&&^
*p* < 0.01 vs. TAT group; ^%^
*p* < 0.05 vs. TAT+AT7 group. (**B**) Statistical results of glioma bioluminescence intensity. (**C**) Body weight changes. (**D**) TUNEL and CD31 staining were performed on glioma tissue sections to observe cell apoptosis and MVD. Bars represent 100 μm. (**E**) Quantification of TUNEL-positive cells. (**F**) Quantification of angiogenesis inhibition rate. Bars represent 100 μm. Each data point represents the mean ± standard deviation. *n* = 3, ** *p* < 0.01, *** *p* < 0.001 vs. AT7 group; ^##^*p* < 0.01, ^###^
*p* < 0.001 vs. TAT group; ^&^
*p* < 0.05, ^&&^
*p* < 0.01 vs. TAT+AT7 group.

## Data Availability

Not applicable.

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
