# Peer review of "A Novel Blood–Brain Barrier-Penetrating and Vascular-Targeting Chimeric Peptide Inhibits Glioma Angiogenesis"

_ijms, 2023, doi:10.3390/ijms24108753_

Round 1

Reviewer 1 Report

The study by Lu L. has been conducted as a follow-up of a previous study published by the same team (Lu L. et al. Int J Nanomedicine 2020). In this previous study, the authors have developed a new tandem peptide TAT-A7 (conjugating TAT and AT7). This peptide has been shown to bind to VEGFR-2 and NRP-1, to be up-taken by HUVEC endothelial cell line, and to penetrate the blood brain barrier and gliomas in rats. In this new study, the authors repeated these results and analyzed the anti-angiogenic and anti-glioma activities of this peptide. They showed that TAT-A7 prevented the binding of VEGF-A to VEGFR2 and NRP-1 and decreased the phosphorylation of different molecules involved in VEGFR2 signaling pathway. TAT-A7 decreased the proliferation and migration capacity of HUVEC, induced their apoptosis. It could also be taken up by a cell line of brain endothelial cells. TAT-A7 had an anti-angiogenic effect in zebrafish embryos, colocalized with endothelial cells in a rat glioma model and seemed to be associated with an increase of apoptosis of tumor cells.

The authors should quantify the colocalization of CD31 and TAT-AT7 (Fig. 9). They should also quantify the TUNEL staining and CD31 staining in Fig 10. Showing only one field is not enough. They should indicate the number of fields they have analyzed on the slides.

The authors should comment on the mechanisms involved in the penetration of the blood brain barrier by TAT-AT7.

Reviewer 2 Report

The manuscript adds to authors' previous study of a TAT-based cell-penetrating peptide conjugate that is active against glioma. The in-vitro experiments, binding and toxicity studies, and in-vivo models were well designed, and the results were well presented. Scope and limitations of the study were also presented in a decent manner.

More analogues could have improved the strength of the manuscript, but the authors focused more on the mechanistic aspects at this time. Overall, the manuscript is a valuable addition to the fields of both cell-penetrating peptides and brain tumor therapeutics. 

Reviewer 3 Report

My main concern is the selectivity of TAT-AT7 to cancer cells vs benign ones.

In Fig.9 the Authors used FITC labeled peptides in vitro without showing the dynamics/distribution in the whole body. Also, in Fig. 10 they demonstrated bioluminescence images visualized by imaging system where bioluminescence corresponds to tumor only. Peptides were not labeled with fluorescent label. As to my opinion this data is missing. Authors should provide an explanation or alternatively do comparative fluorescent examination (at least in vitro) with cancerous vs normal cells from the same origin to show selectivity to cancer cells.

Round 2

Reviewer 1 Report

The authors have improved their manuscript.

Minor point:

Fig 9 : in the figure legend, the authors should explain which parameters they analyzed in panel B

Reviewer 3 Report

No more comments

Author Response

I have replied to the comments.